# Plant Mitochondrial Carriers: Molecular Gatekeepers That Help to Regulate Plant Central Carbon Metabolism

**DOI:** 10.3390/plants9010117

**Published:** 2020-01-17

**Authors:** M. Rey Toleco, Thomas Naake, Youjun Zhang, Joshua L. Heazlewood, Alisdair R. Fernie

**Affiliations:** 1Max-Planck-Institut für Molekulare Pflanzenphysiologie, Am Mühlenberg 1, 14476 Potsdam-Golm, Germany; mtoleco@student.unimelb.edu.au (M.R.T.); Naake@mpimp-golm.mpg.de (T.N.); yozhang@mpimp-golm.mpg.de (Y.Z.); 2School of BioSciences, the University of Melbourne, Victoria 3010, Australia; jheazlewood@unimelb.edu.au; 3Center of Plant Systems Biology and Biotechnology, 4000 Plovdiv, Bulgaria

**Keywords:** MCF, TCA cycle, oxidative phosphorylation, mitochondrial carriers, transporters

## Abstract

The evolution of membrane-bound organelles among eukaryotes led to a highly compartmentalized metabolism. As a compartment of the central carbon metabolism, mitochondria must be connected to the cytosol by molecular gates that facilitate a myriad of cellular processes. Members of the mitochondrial carrier family function to mediate the transport of metabolites across the impermeable inner mitochondrial membrane and, thus, are potentially crucial for metabolic control and regulation. Here, we focus on members of this family that might impact intracellular central plant carbon metabolism. We summarize and review what is currently known about these transporters from in vitro transport assays and *in planta* physiological functions, whenever available. From the biochemical and molecular data, we hypothesize how these relevant transporters might play a role in the shuttling of organic acids in the various flux modes of the TCA cycle. Furthermore, we also review relevant mitochondrial carriers that may be vital in mitochondrial oxidative phosphorylation. Lastly, we survey novel experimental approaches that could possibly extend and/or complement the widely accepted proteoliposome reconstitution approach.

## 1. Introduction

An ancestral endosymbiotic α-proteobacteria fused with an ancestral eukaryotic cell to give rise to the mitochondria approximately 1.5 billion years ago [1]. Since then, metabolic processes involving multiple compartments of the cell are facilitated by specific transmembrane transporter proteins [2]. The intracellular transport of mitochondrial metabolites plays an important role in cellular respiration via the processes of the tricarboxylic acid (TCA) cycle, oxidative phosphorylation, amino acids biosynthesis [3], fatty acids biosynthesis [4], photorespiration [5], and C4 photosynthesis [6]. Metabolic intermediates of these pathways pass through the double membrane of mitochondria that delineates four different sub-compartments: the outer mitochondrial membrane (OMM), the intermembrane space (IMS), the inner mitochondrial membrane (IMM), and the mitochondrial matrix (MM) [7]. The OMM is highly permissive to the passage of ions and small uncharged molecules (<5 kDa) through pore-forming membrane proteins (porins), such as the voltage-dependent anion channels [8]. Larger molecules, especially proteins, must be imported by specialized translocases. By contrast, the IMM is a more stringent molecular barrier allowing only specific metabolites to cross from or into the MM [9]. The highly impermeable IMM is required to establish an electrochemical gradient via the activities of the oxidative phosphorylation membrane protein complexes needed for ATP biosynthesis [10]. Thus, an array of nuclear-encoded mitochondrial carrier (MC) proteins are responsible for the transport of a wide range of substrates shuttled across the IMM. MCs belong to a superfamily of transporters called the mitochondrial carrier family (MCF). Various aspects of plant MCFs have been extensively reviewed by Palmieri et al. (2011) [11], Monné et al. (2019) [3], Haferkamp and Schmitz-Esser (2012) [12], and Lee and Millar (2016) [13].

Most MCF members are relatively small, ranging from 30 to 35 kDa, around 300 amino acids in length, and have a conserved six transmembrane α-helices region [14]. Most of the primary structure of MCs is comprised of three homologous regions, each approximately 100 amino acids in length [15], and both the N and C termini face the IMS [16]. Each repeat region is comprised of two transmembrane segments flanking a short helical region that is oriented parallel to the lipid bilayer [17]. Furthermore, each repeat region is comprised of two hydrophobic transmembrane α-helices connected by a long hydrophilic matrix loop [3,18] and bears the mitochondrial carrier domain superfamily motif (IPR023395). In the odd-numbered α-helices is the conserved PX[DE]XX[RK] motif, the charged residues in this motif can form inter-domain salt bridges (also called matrix salt-bridge network) [19,20,21]. Residues of the salt-bridge form a hydrogen bond with a proximal glutamine residue stabilizing the network (also called Q brace) [19,20,21]. Moreover, in the even-numbered α-helices is another conserved [YF][DE]xx[KR] motif that can form another salt bridge called the cytoplasmic salt-bridge network [19,20,21]. This network is stabilized by the Y brace, hydrogen bonds formed by the tyrosine residue in the motif [19,20,21].

Decades of biochemical and structural data have shed light on the transport mechanism of MCs [21,22]. This transport cycle involves what is now referred to as the alternating access mechanism [20] but was previously described by a number of groups as “gated pore model” [23] or “ping-pong mechanism” [24]. The mechanism describes the process wherein a substrate binds to the transporter in its c-state (cytoplasmic-side open and matrix-side closed state), undergoes a conformational change to a transition state and to an m-state (matrix-side open and cytoplasmic-side closed state), followed by the release of the substrate into the MM [17,20,23,25,26,27]. The counter-substrate binds to the transporter in the m-state, undergoes a conformational change eventually leading to the c-state, and releases the counter-substrate into the cytosol [17,20,23,25,26,27]. Upon release of the counter-substrate, the transporter is ready to begin a new transport cycle. The c-state confirmation was confirmed by the crystal structure of the ADP/ATP carrier in complex with the inhibitor carboxyatractyloside [17]. Recently, an ADP/ATP carrier in its m-state was crystallized in complex with another inhibitor, bongkrekic acid [20]. The transport cycle mechanism of MCFs has recently been reviewed [21,28].

While ADP/ATP carriers have been studied extensively, much is yet to be learned from other members of the family, particularly among plant MCF members. In *Arabidopsis thaliana*, 58 MCF proteins have been identified, but in vitro studies indicate broad substrate specificities and their physiological function *in planta* is largely unknown [11,12,13]. This lack of specificity among plant MCFs is surprising given the metabolic control expected to exist at the IMM. Among the substrates that have been demonstrated to be transported by Arabidopsis MCs across the IMM are: nucleotides and dinucleotides (ATP, ADP, AMP, NAD^+^, FAD/folate); di-/tricarboxylates (malate, succinate, 2-oxoglutarate (2-OG), oxaloacetate (OAA), fumarate, citrate, isocitrate); amino acids (glutamate, aspartate, *S*-adenosylmethionine); cofactors (coenzyme A, thiamine diphosphate); and ions (phosphates, protons, Fe^2+/3+^) [3,12]. Most MC proteins are localized to the IMM, while a few are localized elsewhere. For example, AT5G17400 (ER-ANT) is targeted to the endoplasmic reticulum [29] and some members like AT4G32400 (AtBrittle1) are dual targeted to the mitochondria and plastids [30]. Of the 58 MCF members in *A. thaliana*, only 28 have thus far been confirmed to localize to the mitochondria by organellar proteomics and localization by fluorescent protein tagging, while a total of 12 MCF members have been reported to localize elsewhere [13]. Here, we focus on a subset of MCs that are potentially involved in cellular respiration either in the transport of TCA cycle intermediates or in transport processes relevant to mitochondrial oxidative phosphorylation.

## 2. Mitochondrial Organic Acid Transporters Are Important to Central Carbon Metabolism

Metabolite partitioning between the cytosol and mitochondria, mediated by mitochondrial carriers, is a distinguishing feature of eukaryotic metabolism that necessitates flexibility in the IMM transport direction and kinetics. These transport processes are primarily driven by an electrochemical gradient or proton motive force (PMF), which is comprised of a proton gradient (ΔpH) and membrane potential (ΔΨ) generated across the IMM by proton pumps of the electron transport chain [13]. The location of mitochondrial carriers in three-dimensional space in the context of physical compartmentalization of plant metabolism makes them strategic points for metabolic regulation and control, e.g., in the TCA cycle. In vitro biochemical data regarding mitochondrial organic acid transporters showed that these mitochondrial carriers are most likely responsible for shuttling of TCA cycle intermediates, i.e., between the cytosol and the mitochondria. This strongly suggests that mitochondrial organic acid transporters play a role in central carbon metabolism. However, *in planta* interrogations of the physiological roles of these mitochondrial organic acid transporters are yet to be achieved. To date, there are no reported changes to plant metabolism when mitochondrial carrier loss-of-function plants have been characterized. Recent work on the Arabidopsis TCA cycle interactome shows that a putative phosphate transporter interacted with TCA cycle enzymes [31,32]. However, to date, the physiological significance of these protein–protein interactions remains unknown.

In plants, there are three mitochondrial carriers most likely to be relevant to TCA cycle operation under different flux modes. These are: (1) dicarboxylate carriers (DICs); (2) dicarboxylate/tricarboxylate carriers (DTC); and (3) succinate/fumarate carrier (SFC). To recapitulate the evolutionary relationships among these three transporters in the context of the entire mitochondrial carrier superfamily, we sampled sequences that showed similarity to an amino acid profile of MCF sequences. To this end, we aligned previously known MCF protein sequences and some close paralogs and using MUSCLE [33] built a protein profile using hmmbuild [34] after selecting conserved regions of the alignment via GBLOCKS. The HMM protein profile was queried against the complete proteome files of 69 species to detect protein sequences with similarity to the MCF profile. The resulting matches were aligned (hmmalign) against the protein profile. Homosites with more than 20% missing values, as well as the misaligned N- and C- terminus regions, were removed from the alignment. The phylogenetic relationships were inferred based on Maximum Likelihood using RAxML [35], and branch supports were calculated using BOOSTER [36]. Our analysis showed that DICs, DTC, and SFC are not monophyletic (Figure 1). Mitochondrial organic acid transporter formed two distinct clades. In the first clade, DICs and DTC grouped with 2-OG carriers (OGCs). The SFC formed the second organic acid clade with other non-plant organic acid transporters including oxodicarboxylate carriers (ODCs), citrate carriers (CiCs), and yeast suppressor of HM (histone-like proteins in yeast mitochondria) mutant 2 (YHM2). While these non-plant organic acid transporters likely play a vital role in these species, they will not be discussed in this review. Biochemical data would insinuate that DTC and CiC must be closely related as they both transport citrate; phylogenetic analysis revealed SFC and not DTC, is more similar to CiC (Figure 1). Based on available biochemical data, it appears that transport functions of CiC and DTC have evolved independently but perhaps convergently.

It has been established that the plant TCA cycle can also operate distinctly from the textbook cyclic mode [13,37] (Figure 2). The well-established cyclic mode of TCA flux most often associated with non-photosynthetic organisms is most likely to operate in leaves in the dark when there is a high demand for ATP through cellular respiration (Figure 2A) [13]. Import of pyruvate may be exclusively attributed to the mitochondrial pyruvate carriers (MPCs) [38]. However, malate/2-OG exchange could be undertaken by either DICs and/or DTC. On the other hand, fumarate efflux is probably catalyzed by SFC using 2-OG as counter-substrate as there is no net flux of succinate reported. The observation that these different non-cyclic modes are dictated by cellular metabolic demands suggests some level of control. However, whether regulation of the activity and/or expression of these transporters exist is still an open question. Based on the available biochemical data on these relevant transporters (see below), we can begin to put forward some theories regarding their potential roles in central carbon metabolism. Several metabolites associated with the TCA cycle have been proposed to exchange across the IMM and thereby link the operation of several enzymes in mitochondrial to those in other cellular compartments. For example, citrate is suggested to be exported to the cytosol and then converted to 2-OG for redistribution to either chloroplasts or mitochondria [39]. Similarly, the malate-OAA shuttle can possibly mediate photorespiration through the mitochondrial malate dehydrogenase (mMDH) catalyzed reversible reaction after OAA import into the mitochondria [39,40].

During the day, ATP demand from cellular respiration is low and enzyme kinetic analysis suggests that this results in the operation of a non-cyclic flux mode [37] (Figure 2). In this model, pyruvate entry into the TCA is reduced since the pyruvate dehydrogenase complex is inactivated in the light by phosphorylation. The model, furthermore, predicts that there is a net influx of malate accompanied by a net efflux of citrate, a metabolite exchange that is in congruence with the in vitro transport activity of DTC. Net import of OAA by the mitochondria during the day has been proposed on the basis of experiments described in the literature [13]. Such a net influx of OAA could also be mediated by DTC in exchange with citrate. We propose that DTC is sufficient to support this flux, partly explaining the very high DTC protein abundance in an average Arabidopsis mitochondrion [43] (Figure 2B). A non-cyclic flux mode of the TCA cycle has been proposed based on evidence from isotope labeling studies in illuminated leaves of *Xanthium strumarium* [42]. Here, there is a net influx of OAA as well as citrate. However, this influx has also proposed to be accompanied by net efflux of malate, fumarate, and 2-OG. In this model, the metabolite fluxes can only be supported by the concerted activities of DICs, DTC, and SFC. Citrate influx is mediated by DTC in exchange with OAA, malate, or 2-OG, while fumarate is transported by SFC using OAA or 2-OG as counter-substrate. DICs may also play a role in the transport of malate, 2-OG, and OAA. Alternatively, stored citrate can be converted to 2-OG via isocitrate in the cytosol [37]. In this case, the remaining metabolic fluxes can be attributed to the activities of DICs and SFC. The overlap of substrates allowed for transport by these transporters makes it challenging to pin down the responsible transporter for a specific metabolite exchange. While absolute quantification of the subcellular levels of organic acids remains a major analytical challenge, there is an acute need to define the *in planta* transport substrates and the directionality of the transport carried out by these MCs in order to complete our understanding of mitochondrial metabolism as they potentially dictate the metabolic fates of the TCA intermediates.

### 2.1. Dicarboxylate Carriers (DICs)

DICs are members of the MCF reported to facilitate the transport of dicarboxylates such as malate and related compounds as well as phosphate, sulfate, and thiosulfate across the IMM [44]. Phylogenetic analysis showed that distinct DIC kingdom-level subclades can be distinguished clearly, separating those of fungal, animal, or plant origins (Figure 1). All higher land plants in our analysis possess at least one copy of DIC. However, there is no DIC homolog in algae and it seems to have first evolved in bryophytes (*Selaginella moellendorffii* and *Physcomitrella patens*), which have three and five DIC homologs, respectively. This is consistent with the analysis of MCFs in *Ostreococcus lucimarinus* where a DIC homolog could not be identified [11]. DIC gene duplication seemed to be more common in plants than in animals. Consistent with our analysis, *A. thaliana* was reported to have three DIC homologs—AtDIC1 (AT2G22500), AtDIC2 (AT4G24570), and AtDIC3 (AT5G09470) [44]. *Populus trichocarpa* has eight DICs that all belong to the AtDIC2 subgroup. Cassava (*Manihot esculenta*) possesses five DICs, clustering with the AtDIC2 subgroup, and two additional DICs in the AtDIC3 subgroup.

Arabidopsis DICs have been characterized in vitro and were shown to have varying transport kinetics but similar substrate selectivity, transporting mainly malate, OAA, succinate, maleate, malonate, phosphate, sulfate, and thiosulfate [44]. In the case of malate homo exchange, the V_max_ for AtDIC3 (2.21 ± 0.31 mmol min^−1^ g protein^−1^) was at least double of AtDIC2 (1.01 ± 0.11 mmol min^−1^ g protein^−1^) and at least seven times compared to that of AtDIC1 (0.29 ± 0.06 mmol min^−1^ g protein^−1^). While AtDIC3 seems to be the most efficient transporter for malate, its transcript could not be detected in any of the tissues tested [44]. Recent Arabidopsis mitochondrial proteomic surveys support the observation that AtDIC3 does not appear to be highly expressed [43]. By contrast, AtDIC1 (59 protein copies per mitochondria) was found to be slightly more abundant than AtDIC2 (21 protein copies per mitochondria) [43]. However, transcript levels of *AtDIC1* and *2* appear to vary across tissues [44]. The mRNA level of the former was found to be higher in roots and in flowers, while the latter was more abundant in leaves, stems, and seedlings. It should be noted that the precise roles of the AtDIC homologs remain to be investigated *in planta*. Thus, it is still unknown whether these homologs are functionally redundant or if indeed they can transport such a wide array of substrates. Furthermore, *K*_m_ values were reported to be in the millimolar range, e.g., 0.4 ± 0.09 mM for AtDIC1 during malate/malate exchange [44] as opposed to the ADP/ATP carrier (AtAAC1) that was shown to be in the millimolar range [45]. The orders of magnitude of the *K*_m_ and *V*_max_ values are similar to ones measured for DIC homologs in *Drosophila* [46] rat and *C. elegans* [47]. We hypothesize that this might be a form of metabolic control, i.e., metabolites are exported/imported only when a certain threshold concentration is achieved. For example, malate synthesized in the mitochondria is not exported as soon as it is formed by the action of DICs, rather, only when a critical concentration is reached would DICs transport the metabolites in excess. The transport process is also rapid as reflected by the *V*_max_ values consistent with our hypothesis of *relief* from metabolite accumulation.

### 2.2. Dicarboxylate/Tricarboxylate Carrier (DTC)

DTCs facilitate the transport of dicarboxylates such as malate and 2-OG and tricarboxylates such as citrate [48]. Our phylogenetic analysis revealed that DTC homologs are present in all included plant species and in some fungal species (Figure 1). DTCs and OGCs share a common ancestor. The relative position of the two subgroups indicates that the ancestral gene is closely related to DICs. We hypothesize that the common ancestral gene was duplicated and underwent neofunctionalization upon speciation. The animal copy became OGC while in plants it became DTC. Alternatively, neofunctionalization happened in the last common ancestor and the plant and animal lineages received DTC and OGC, respectively, upon speciation. Hints to support the second hypothesis are provided by the proteomes of some protists. There are three species that contain both OGC-like and DTC-like proteins: *Aureococcus anophagefferens* (Class Pelagophyceae), *Emiliania huxleyi* (Class Prymnesiophyceae), and *Tetrahymena thermophila* (Class Oligohymenophorea). However, other protists possess only OGC-like transporters as in *Leishmania major* and *Trypanosoma brucei* and yet other protists, such as *Plasmodium falciparum*, only have DTC-like transporters.

DTCs are among the most abundant MC proteins in the Arabidopsis IMM comprising 0.8% of the total IMM area, i.e., 6836 protein copies per mitochondria [43]. Unlike the other three highly abundant MC proteins: ADP/ATP carriers (AtAAC1-3; 6.2% of IMM area; 53,065 protein copies/mitochondria); mitochondrial phosphate carriers (AtMPT2-3; 2.5% of IMM area; 21,325 protein copies/mitochondria); and, uncoupling proteins (AtUCP1-3, 1.0% of IMM area; 8595 protein copies/mitochondria), there is only one DTC homolog in Arabidopsis. DTCs have been studied in a few plants including Arabidopsis and *Nicotiana tabacum* [48], *Vitis vinifera* (grapes) [49], *Helianthus tuberosus* (Jerusalem artichoke) [50], and *Citrus junos* (yuzo) [51]. The purification and characterization of a citrate transporter in maize have been described [52]. The reported activity may represent the MzDTC homolog since the transport substrates of the maize citrate transporter closely resemble that of AtDTC. In the plant kingdom, the numbers of DTC homologs vary without any clear pattern. A single homolog was found in *Chlamydomonas reinhardtii*, while there are two and three homologs in the mosses *S. moellendorffii* and *P. patens*, respectively. In the *Brassica* genus, the number of DTC homologs varies from one in *A. thaliana*, Arabidopsis *lyrata*, and *Capsella rubella*, two in *Brassica oleracea*, and three in *Brassica rapa*. *N. tabacum* has four homologs (NtDTC1-4) consistent with that reported in literature [48]. Regalado et al. (2013) [49] reported that there were three DTC homologs in *V. vinifera* (VvDTC1-3) and that VvDTC2 and VvDTC3 reached high transcript levels in the berry mesocarp at the onset of ripening. However, the phylogenetic analysis here showed that only two of these homologs clustered with plant DTCs. Our analysis revealed that VvDTC1 (vitvi_GSVIVT01025463001) is similar to AtDTC. An investigation of VvDTC2 and VvDTC3 revealed that these two proteins have the same locus tag indicating they are likely the same gene, and this would correspond to the second VvDTC observed in our analysis. Finally, a single DTC homolog was cloned from *C. junos* [51] and, similarly, we were able to detect only one DTC homolog in the close relative *Citrus sinensis*.

AtDTC and NtDTCs have been assigned a transport function that involves an obligatory electroneutral exchange of dicarboxylates such as malate and 2-OG and tricarboxylates such as citrate [48]. It was demonstrated that DTCs were able to catalyze homoexchange transport, i.e., dicarboxylate/dicarboxylate and tricarboxylate/tricarboxylate on top of the dicarboxylate/tricarboxylate transport modality. It remains to be seen, however, which of these modalities are relevant *in planta*. It is clear from the in vitro transport data [48] that DTCs are promiscuous in terms of transport substrate. DTCs can transport almost all the intermediates of the TCA cycle except fumarate and succinyl-CoA for which there is no available data. For AtDTC, the homoexchange kinetic constants measured for different substrates in two different pH values showed that regardless of the substrate, the *K*_m_ and V_max_ changed as a function of pH. The *K*_m_ values generally increased at pH 7, indicating that substrate affinities were decreased; V_max_ values were also decreased at pH 7. These changes in the transport kinetics as a function of pH is critical because it has been shown that in Arabidopsis, the pH of the MM is slightly basic (pH 8.1) and that cytosolic pH is close to neutral, pH 7.3 [53]. It is, however, important to note that these in vitro data were not obtained under conditions representative of physiological conditions. To our knowledge, there are no reports where the external and the internal pH mimic physiological values in DTC transport measurements.

### 2.3. Succinate/Fumarate Carriers (SFC)

SFCs facilitate the exchange of succinate and fumarate [54]. In plants, SFC is present in all lineages, in the green alga, *Chlamydomonas*, but not in the red alga *Cyanidioschyzon merolae*. Our analysis also indicates that animals do not have the SFC homolog.

The gene encoding SFC was first described in the yeast mutant *arc1* while screening for a mutant unable to utilize ethanol as the sole carbon source [55]. The yeast SFC was shown to transport fumarate, succinate, methylfumarate, 2-OG, and OAA against [^14^C]oxoglutarate [54]. SFC was further shown to prefer succinate and fumarate as substrates since the presence of either substrate almost completely inhibits fumarate/[^14^C]oxoglutarate exchange [54]. Catoni et al. (2003) [56] complemented the *sfc* yeast mutant (*arc1*) with an Arabidopsis SFC homolog (AT5G01340) that was 35% similar to the ScSFC gene. This resulted in the re-establishment of the yeast to grow in minimal media with ethanol as the sole carbon source. However, the transport behavior of AtSFC is yet to be described.

In yeast, SFC may play a role in shuttling cytosolic succinate from the glyoxylate cycle [54,57,58]. The transported mitochondrial fumarate can be acted upon by cytosolic fumarase to yield malate that could ultimately be used for gluconeogenesis [54,57]. Thus, SFC may potentially link TCA, glyoxylate cycle, and gluconeogenesis [56]. In plants, the glyoxylate cycle is needed in lipid mobilization especially during the early stages of germination. The β-glucuronidase (GUS) reporter system has been used to show that the SFC promoter was active in the cotyledons, hypocotyls, and root tips consistent with the hypothesized role in early germination [56]. The *At**SFC* promoter was also active in pollen and during in vitro germination of pollen tubes, which is also consistent with upregulation of glyoxylate cycle genes during pollen development. Catoni et al. (2003) [56] also reported that the *AtSFC* promoter is active in specific regions of mature leaves, i.e., patches of veins and trichomes. Moreover, a recent report on the Arabidopsis mitochondrial proteome showed that an average mitochondrion has only 73 SFC protein copies, which are around the same number as DIC1 and DIC2 combined [43]. Based on the TCA flux mode during the day inferred from enzyme kinetic analysis (Figure 2B), SFC is unlikely to play a role in TCA metabolite shuttling. However, TCA flux model based on isotope labeling experiments (Figure 2C) and during the night (Figure 2A) indicated that there is a net efflux of fumarate from the mitochondria. We can hypothesize that perhaps SFC plays a role in this transport process since DICs and DTC are not able to efficiently transport fumarate [44,48]. It has not escaped our attention that there has been no report or model prediction showing net influx of succinate to the mitochondria, which would have been expected as succinate is the preferred substrate of non-plant SFCs. It is thus likely that SFC is using another metabolite as a counter-substrate to facilitate fumarate transport, possibly OAA and 2-oxogulatrate, in congruence with yeast SFC transport assays [54].

## 3. Mitochondrial Carriers Relevant to Mitochondrial Oxidative Phosphorylation

Oxidative phosphorylation (electron transport-linked phosphorylation) is the route by which ATP is formed as a result of the transfer of electrons from NADH or FADH_2_ to O_2_ by a series of electron carriers in the IMM [59]. The oxidative phosphorylation system is composed of the mitochondrial electron transport chain (ETC) and the ATP synthase complex (Complex V) from bacteria to higher eukaryotes [59]. To drive favorable translocation of charged species across the IMM, the ΔΨm and ΔpH generated through the action of the mitochondrial respiratory electron transfer chain could influence the transporter activity. The electrochemical gradient is ultimately used by the ATP synthase to produce ATP. Therefore, the state of oxidative phosphorylation would be expected to affect the kinetic behavior, specificity, transport orientation, and cooperativity of IMM transporters. There are three transport processes directly or indirectly related to mitochondrial oxidative phosphorylation that is mediated by MCs: (1) transport of NAD^+^, (2) proton translocation, and (3) transport of adenine nucleotides (Figure 3).

### 3.1. NAD^+^ Transport

As a coenzyme for redox processes, NAD^+^ is playing important roles in the operation of a wide range of dehydrogenase activities, signaling pathways through their interaction with reactive oxygen species (ROS) and generation of NADH from oxidative phosphorylation. NAD^+^ is essential for several metabolic pathways including glycolysis, TCA cycle, glycine decarboxylation, the Calvin–Benson cycle, and β-oxidation in peroxisomes [62]. Hence, the movement of NAD^+^ from different subcellular compartments is mediated by different subcellular NAD^+^ transporters. In plants, both de novo and salvage NAD^+^ biosynthetic pathways culminate in the synthesis of nicotinate mononucleotide (NaMN) [63]. The salvage pathway starts with nicotinamide (NAM) or nicotinic acid (NA), while the de novo pathway starts in plastids using aspartate or tryptophan as precursors. Both metabolic fluxes converge in the formation of nicotinic acid mononucleotide (NAMN), which, in turn, gives rise to NAD^+^. Since the last step of NAD^+^ synthesis takes place in the cytosol, NAD^+^ must be imported into the mitochondria to allow TCA cycle metabolism and oxidative phosphorylation [64]. In Arabidopsis, there are three MCF members responsible for NAD^+^ transport, AtNDT1 (AT2G47490) and AtNDT2 (AT1G25380), targeted to the IMM, and AtPXN (AT2G39970), located in the peroxisomal membrane [64,65]. Although previous research suggested that AtNDT1 is targeted to the inner membrane of chloroplasts [65], recent subcellular localization experiments and proteomics data revealed that AtNDT1 locates exclusively to the IMM [64]. Both AtNDT1 and AtNAD2 are able to complement the phenotype of a yeast mutant lacking NAD^+^ transport [65]. Interestingly, both AtNDT1 and AtNDT2 have similar substrate specificity; importing NAD^+^ against ADP or AMP; they do not accept NADH, nicotinamide, nicotinic acid, NADP^+^, or NADPH as transport substrates [65]. The AtPXN transporter has a more versatile transport behavior, able to accept NAD^+^, NADH, and CoA in vitro [66,67,68]. In addition, as a pyridine nucleotide, NAD^+^ is involved in the transport of electrons within oxidation–reduction reactions as well as being a highly important component of cellular signaling [63]. Given that the redox status regulates the plant TCA cycle [69], NAD^+^ import not only provides co-enzymes but might also act as a signal that regulates central metabolism.

### 3.2. Uncoupling Proteins (UCPs)

Uncoupling proteins (UCPs) are suggested to mediate a non-phosphorylating free fatty acid-activated proton re-entry into the MM leading to a thermogenic dissipation of proton gradients, thereby uncoupling oxidative phosphorylation [61,70]. In animals, the protonophoretic function of UCP1 in brown adipose tissues leading to thermogenesis is of great importance among newborns, cold-acclimated, and hibernating mammals [71,72]. While initial reports in Arabidopsis indicated the presence of 6 UCPs [73], current studies consider the genome to encode three UCP homologs (UCP1-3: AT3G54110, AT5G58970, and AT1G14140) [11]. AtUCP1 was shown to be needed for efficient photosynthesis [60]. During photorespiration, the mitochondrial conversion of glycine to serine by glycine decarboxylase leads to an accumulation of NADH. Subsequently, the NADH is used by MDH or is funneled to the electron transport chain. Since there is a lower demand for mitochondria-derived ATP during the day, photorespiration leads to a substantial increase in the proton gradient by virtue of increased NADH oxidation in Complex I. Without ATP synthesis-coupled proton re-entry into the MM, UCPs may thus provide a mechanism to dissipate proton gradient build-up so as not to restrict electron flow to regenerate NAD^+^ needed in photorespiration [60]. This hypothesis is supported by the observation that *ucp1* mutants showed a decreased photorespiratory flux from glycine to serine. However, there was no concomitant accumulation of glycine accumulation in *ucp1* mutants. It was suggested that there might be a slight overcompensation in upstream photorespiratory regulatory mechanisms [60]. UCPs are known to curb mitochondrial reactive oxygen species production, which was observed in *ucp1* mutants alongside a significant decrease in the activities of two mitochondrial enzymes, malic enzyme and aconitase, both are particularly sensitive to oxidative inactivation [60]. It should, however, be noted that proton transport in this study was not directly measured. For example, 6-methoxy-*N*-(3-sulfopropyl)-quinolinium (SPQ) can be used to monitor proton flux and has been used for the functional characterization of UCP homologs [74]. Recently, the function of AtUCP1 and 2 has been reported as the transporter of amino acids (aspartate, glutamate, cysteine sulfinate, and cysteate), dicarboxylates (malate, OAA, and 2-OG), phosphate, sulfate, and thiosulfate [61]. The function of AtUCP1 and AtUCP2 is also suggested to catalyze an aspartate out/glutamate in exchange across the mitochondrial membrane and, thereby, contribute to the export of reducing equivalents from the mitochondria in photorespiration [61]. AtUCP1 and AtUCP2 thus have very broad substrate specificities compared to most MCs thus far characterized [61], especially the dicarboxylates of TCA cycle. To reconcile with the earlier proposal regarding the role of AtUCP1 in photorespiration [60], it has been suggested that the role of AtUCP1 and AtUCP2 may be in the glycolate pathway for the shuttling of redox equivalents across the mitochondria as part of the malate/aspartate shuttle [61].

### 3.3. Adenylate Transporters

In plant mitochondria, a suite of proteins transports various forms of adenine nucleotides. These include ADP/ATP carriers (AAC1-3: AT3G08580, AT5G13490, AT4G28390), adenedine nucleotide transporter (ADNT1: AT4G01100), adenine nucleotide/phosphate carriers (APC1-3: AT5G61810, AT5G51050, AT5G07320), and AtBrittle1 (AT4G32400). Of these, AAC1-3 occupy 6.2% of the total IMM surface area, and together with APCs and ADNT1 comprise 54,114 proteins copies of the total 1,390,777 proteins in the mitochondrial proteome [43].

Our current understanding of MCFs is mostly due to work done on ADP/ATP carriers [20,75]. Nonetheless, it is still surprising that there is only one report on a plant ADP/ATP carrier homolog, which could be explained by their presumed functional redundancies. AtAAC1 and AtAAC2 were functionally expressed and inserted in the *E. coli* cellular membrane and both preferentially export ATP [45]. However, to date, there is no *in planta* work undertaken on this set of plant transporters. The expression of ADP/ATP carriers can vary in response to various stresses as analyzed from transcriptomic datasets [76]. For example, *AtAAC3* is upregulated in roots under osmotic, salt, oxidative, heat, UV-B irradiation, and mechanical stresses. On the other hand, *AtAAC1* expression remained invariant [76]. This suggests differential transcriptional regulation of the various ADP/ATP carriers, especially under stress.

Functional characterization of AtAPCs indicated that unlike their closest human orthologs, these adenylate transporters do not show a preference for Mg-ATP as a substrate [77,78,79]. AtAPCs were able to accept inorganic phosphate, AMP, ADP, ATP, or adenosine 5′-phosphosulfate as well as ATP in complex with metal divalent ions (Ca^2+^, Mn^2+^, Fe^2+^, and Zn^2+^) as a transport substrate, and, interestingly, AtAPCs were not completely inhibited by either carboxyatractyloside or bongkrekic acid, both of which are powerful inhibitors of ADP/ATP carriers [22,77,79]. The transport activities of AtAPCs were shown to be Ca^2+^-dependent, probably due to the presence of a calcium-ion binding motif in the N-terminal [77,78,79]. Thus, the transporter activities were abolished in the presence of chelating agents such as EDTA and EGTA [77,78]. Since calcium is an important plant signaling molecule, it is possible that mitochondrial energy metabolism under stress is mediated by regulating expression of *AtAPCs* via Ca^2+^. Transcriptomic datasets have shown that *AtAPC1* is highly induced under drought, salt, or osmotic stress [76].

AtADNT1 was described to facilitate the exchange of AMP and ATP [80]. GUS expression reporter assays showed that the *AtADNT1* promoter was highly active in seedling roots, radicles, cotyledon vascular tissues, and in leaf primordia [80]. Furthermore, lower promoter activity was observed in the leaves of adult plants but higher promoter activity in the vascular tissues of sepals. The physiological role of AtADNT1 is yet to be elucidated. However, it was observed that *adnt1* mutants had significantly reduced root growth rates accompanied by impaired root respiration [80]. It was hypothesized that AtADNT1 is relevant in the mitochondrial uptake of AMP especially during exposure to stress such as hypoxia whereupon AMP tend to accumulate in the cytosol [80]. However, *AtADNT1* transcripts were not markedly increased during hypoxic, cold, heat, drought, salt, or oxidative stresses [76]. It was, however, noted that the expression of *AtADNT1* was upregulated during senescence [76,80]

AtBrittle1 (AtBt1) localizes to the endoplasmic reticulum when transiently expressed in Arabidopsis protoplasts [81] but localized to chloroplasts when transiently expressed in tobacco leaf protoplasts [82]. Finally, when fused with GFP and stably expressed in Arabidopsis, it was found to be dual targeted to both mitochondria and chloroplasts [30]. GUS reporter expression assays showed that *AtBt1* was highly expressed in root tips and germinating pollen [82]. When expressed in intact *E. coli* cells, AtBt1 facilitated a unidirectional uptake of AMP, ADP, and ATP [82]. It was hypothesized that AtBt1 is crucial for shuttling newly synthesized adenylates from the plastids [82]. Homozygous *atbt1* lines are highly compromised, and if they survive they do not produce fertile seeds [82]. This suggests that the physiological role of AtBt1 is not compensated by the other adenylate transporters in its absence. Bahaji et al. (2011) [83] showed that when AtBt1 was targeted specifically to the mitochondria, the construct was able to rescue the growth and sterility phenotypes of *atbt1* mutants, indicating that the mitochondrial role of AtBt1 is important for normal growth and development through a mechanism yet to be understood.

## 4. Novel Approaches to Investigate MCFs

The most widely used system for studying the transport properties of MCFs is through expression–purification–reconstitution assays [18]. In this approach, the candidate gene is cloned and homologously [84] or heterologously expressed in either *E. coli* [85,86], *Lactococcus lactis* [87], or yeast [85,87]. The overexpressed protein is isolated and reconstituted into liposomes and the transport activity is measured by direct uptake or export of radioactively labeled substrates. Although the assay seems straightforward, its execution is technically challenging. Much of the current knowledge regarding MCF members was gained through the expression–purification–reconstitution assay system. One limitation of the expression–purification–reconstitution assay is the use of radiolabeled substrates. Aside from the usual prohibitive costs, risks, and regulatory hurdles associated with the use of radioactive materials, the range of substrates that can be directly tested is severely hampered by the availability of radioactive substrates. Most MCF research groups circumvent this obstacle by co-incubating the radioactive substrate with the unlabeled candidate substrate and measure the inhibitory effect compared to the assay measurements without the competing substrate. While not a direct evidence of transport, this approach show competition for the binding site. For example, the substrates for 3′-phosphoadenosine 5′-phosphosulfate Transporter 1 and 2 (PASPT1 [85] and PAPST2 [86]) were assessed in this manner. Still, the better way of doing the transport assay is to directly monitor the uptake of the labeled substrate. To overcome issues associated with the use of radioactive substrates, Rautengarten and colleagues [88] have developed a transport assay system that combines liposome reconstitution and mass spectrometry to directly analyze transport properties of nucleotide sugar transporters in Arabidopsis. Theoretically, this system could be used to study the transport behavior of MCF proteins. Monne and colleagues [79] have already demonstrated the use of inductively coupled plasma mass spectrometry to measure divalent ions to characterize APCs from Arabidopsis and humans.

Protein-effector thermostability shift assays were recently introduced to screen for potential substrates. This is based on the principle that physiologically meaningful interactions result in more stable configurations. Mechanistic insights into the interaction of MC proteins with their substrates, inhibitors, and lipids can be evaluated by monitoring their thermostability [89]. While not direct evidence of transport, the results give a strong indication of the binding of a potential substrate, inhibitor, or other effectors. The assay uses a thiol-specific fluorochrome *N*-[4-(7-diethylamino-4-methyl-3-coumarinyl) phenylmaleimide (CPM), which forms fluorescent adducts with cysteine residues in the protein. As the protein is thermally denatured, the buried cysteine residues become available to CPM, which can be taken as a readout for the denaturation process [90]. Thus, shifts in the thermostability of the protein in the presence of an effector molecule can be measured fluorometrically. The approach has been extended to widen the chemical search space for screening potential MCF substrates that would otherwise be largely delimited by using radioactive substrates [91].

While not members of the MCF, recent work on yeast mitochondrial pyruvate carriers (MPCs) showed that by mimicking the physiological pH gradient between the mitochondria and the cytosol has resulted in quantifiable pyruvate transport [92]. In the absence of the pH gradient, no transport can be detected [92]. This could also be true among MCF members, and, thus, it may be worthwhile to run the MCF transport assays under physiological pH conditions. Perhaps the transporter activity of the MCFs proteins can also be modulated by forming complexes with interacting partner/s. The glutathione transport function of OGC could not be replicated using the typical expression–purification–reconstitution assay approach. An anti-apoptotic protein, Bcl-2, was found to be an interacting protein partner of rat OGC and when co-expressed with OGC in CHO cells, the total mitochondrial glutathione content was significantly increased 24 h post-transfection [93].

## 5. Perspectives

The transport activities of many plant MCs have been characterized in vitro [11,12]; however, most are yet to be interrogated *in planta*. Specifically, transporters such as DICs, SFC, and DTC, which have been shown to transport TCA cycle intermediates in vitro, have not had their physiological functions clearly elucidated *in planta*. A common feature is their broad substrate specificities in vitro. Whether a more stringent gating mechanism exists *in planta* is one of the more interesting questions. Thus far, there is no experimental evidence regarding how these transporters might impact cellular respiration, carbon metabolism, or cellular redox poise. For example, 2-OG shuttling is critical as it integrates carbon and nitrogen metabolism [39,40], while data indicates that some MCs interact with TCA cycle enzymes [31,32]. To date, the physiological significance of these potential protein–protein interactions is yet to be elucidated. Perhaps, these MCs participate in the formation of TCA metabolon serving as membrane anchors. Data from yeast MPCs suggested that formation of complexes modulate transporter activity. Whether canonical MCF members’ activity can also be modulated by the formation of protein complexes and by extension, impose a stricter substrate specificity is an attractive area of research. Recently, it was shown that the transport activity of the human CiC seemed to be modulated via acetylation of a lysine residue in response to glucose supply [94]. Moreover, the transport activity of the rat carnitine/acylcarnitine carrier was shown to be modulated by glutathionylation [95]. It will be interesting to ascertain how other post-translational modifications might modulate MC transport activity and whether this can also be observed *in planta*. More recently, it was shown the circadian protein CLOCK (Circadian Locomotor Output Cycles Kaput) was found to bind the human and mouse DIC, suggesting that a possible connection between circadian rhythm and mitochondrial metabolism may be mediated by dicarboxylate carriers [96]. To date, there have been no such reports on the connection between plant circadian rhythm and plant MCs and how this might be relevant in the diurnal regulation of plant metabolism. While these questions are no doubt interesting, the major challenge in our understanding of the function of these carriers is presented by their apparent functional redundancy. This feature renders it difficult to tease apart their physiological function—the broader adoption of CRISPR/Cas9 based approaches in plants [97] and natural variance screening methods [98] may address this challenge. That said, we still lack biochemical data for many of the members of the plant MCF. It is likely that only as a result of data coming from multiple approaches, we will be able to fully comprehend their physiological importance in the regulation of plant central carbon metabolism.

## Figures and Tables

**Figure 1 plants-09-00117-f001:**
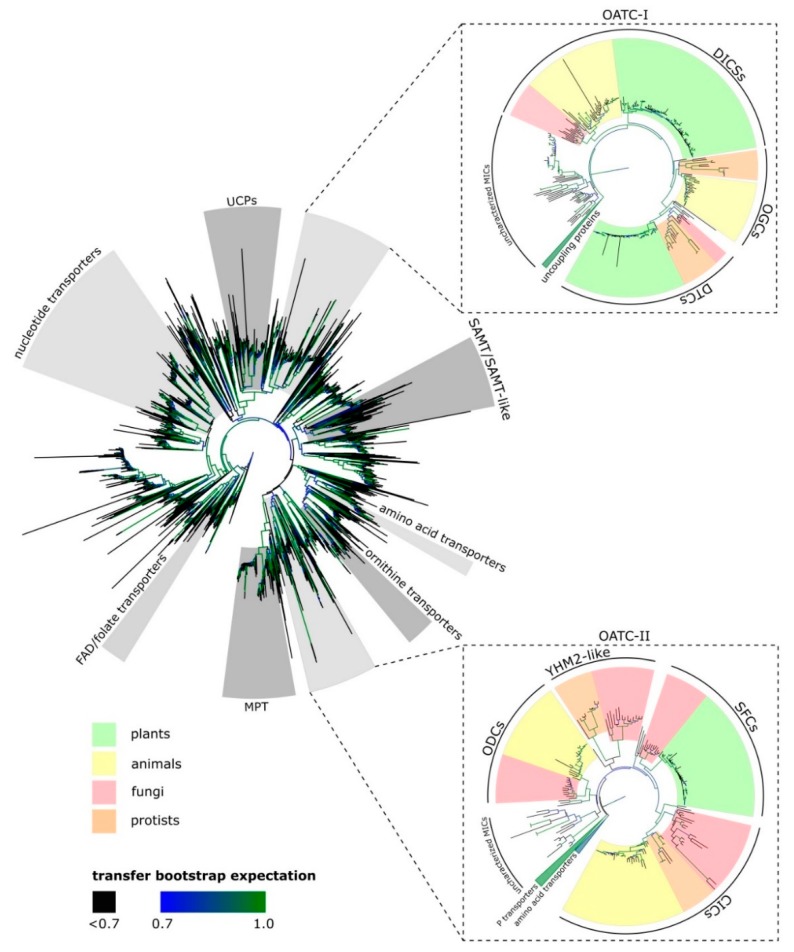
An unrooted phylogenetic tree of mitochondrial carrier families (MCFs) across Domain Eukaryota. The tree was inferred based on Maximum Likelihood using the RAxML software and visualized using iTOL (https://itol.embl.de/), see text for details. UCP: uncoupling proteins, SAMT: S-adenosyl methionine transporter, OATC-I/II: organic acid transporters clade I/II, DICs: dicarboxylate transporters, OGCs: 2-oxoglutarate carriers, DTCs: dicarboxylate/tricarboxylate carriers, ODC: oxodicarboxylate carriers, YHM2-like: yeast HM mutant 2-like transporters, SFCs: succinate/fumarate carriers, CiCs: citrate transporters.

**Figure 2 plants-09-00117-f002:**
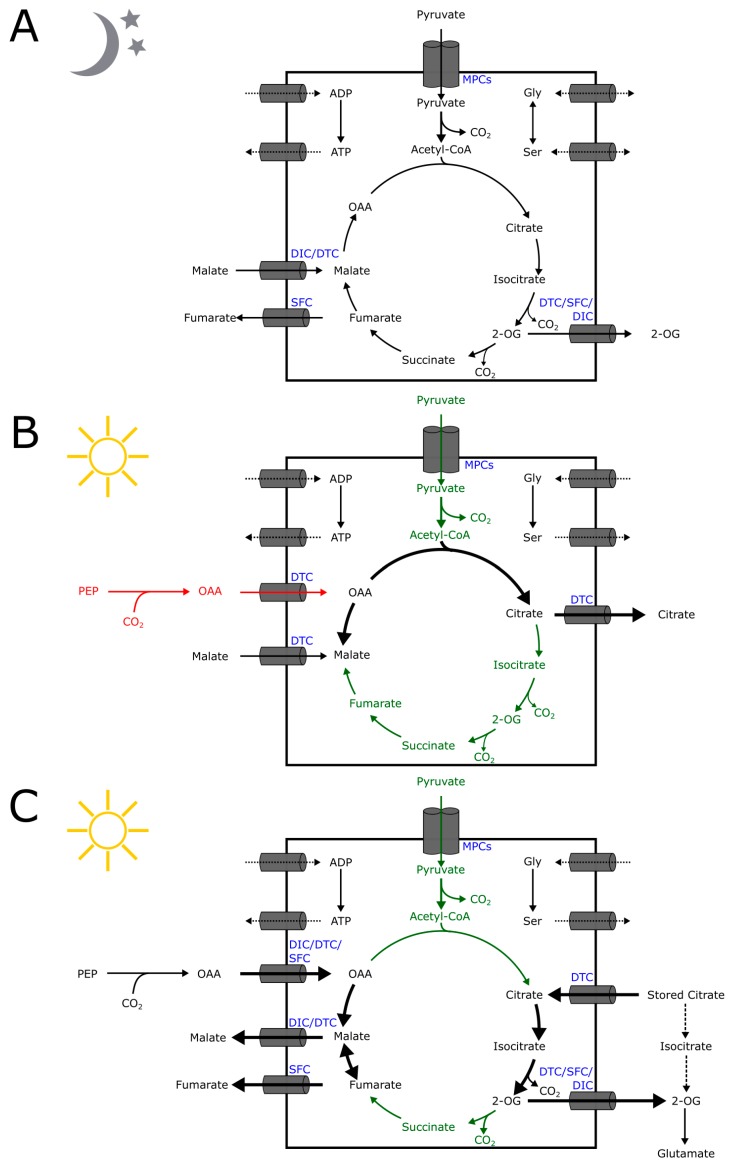
The plant central metabolism related to the mitochondrial Organic Acid Transporters (**A**) In the dark, the TCA cycle most likely operate in the familiar cyclic mode. (**B**) Non-cyclic TCA flux mode has been modeled based on enzymatic kinetic analysis during the day [37,41]; flux in red is based on non-modeling literature. (**C**) A non-cyclic flux mode of the TCA cycle based on isotope labeling studies [42], an alternative metabolic route of stored citrate is shown as broken black arrows. Fluxes shown in green are either inactive or significantly reduced.

**Figure 3 plants-09-00117-f003:**
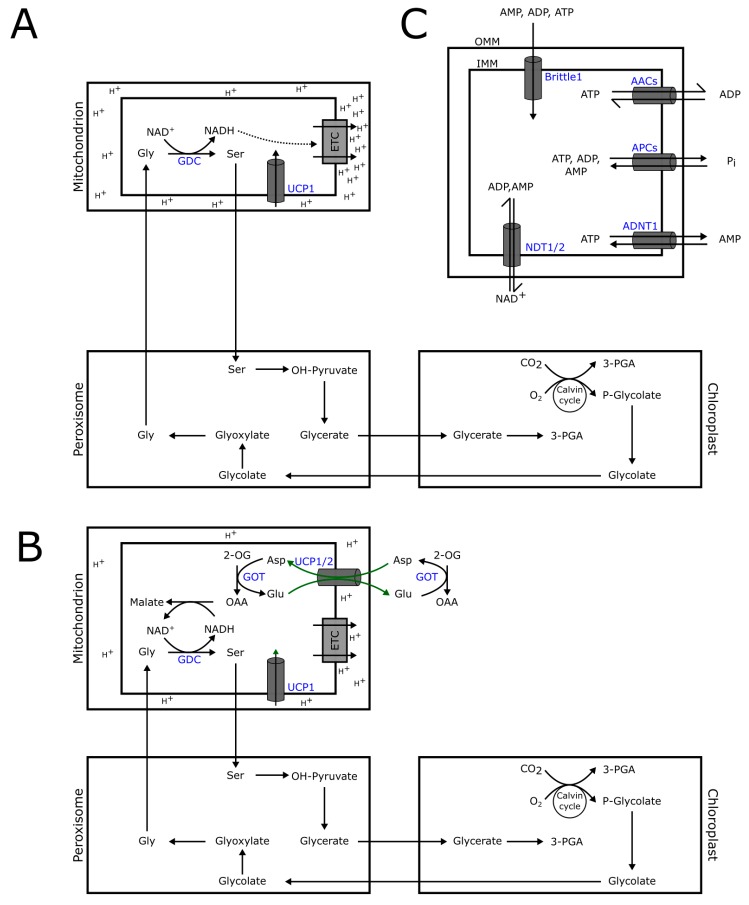
Mitochondrial carriers relevant to mitochondrial oxidative phosphorylation. (**A**) Sweetlove et al. [60] proposed that the function of UCP1 in photorespiration is in the dissipation of protons generated by the increased flux of NADH to the complex 1 of the electron transport chain (ETC). (**B**) Monne et al. [61] showed that UCP1 and 2 mediate aspartate/glutamate exchange. It was proposed that UCP1/2 play a role in dissipating reducing equivalents across the mitochondrial membrane as part of the malate/aspartate shuttle during photorespiration. (**C**) NDT1 and 2 facilitate the transport of NAD^+^ from the cytosol in exchange with ADP or AMP from the mitochondria. ADP/ATP carriers, APCs, ADNT1, and AtBrittle1 mediate the transport of adenine nucleotides. UCP1/2: uncoupling protein 1/2, GDC: glycine decarboxylase complex, GOT: glutamic oxaloacetic transaminase, AACs: ADP/ATP carriers, APCs: ATP/Pi carriers, ADNT1: adenine nucleotide transporter 1, NDT1/2: NAD^+^ transporter 1/2.

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
