# Peer review of "Plant Mitochondrial Carriers: Molecular Gatekeepers That Help to Regulate Plant Central Carbon Metabolism"

_plants, 2020, doi:10.3390/plants9010117_

Round 1
Reviewer 1 Report
Plant Mitochondrial Carriers: Molecular Gatekeepers that Help to Regulate Plant Central Carbon Metabolism
This is a well written and insightful review of the role of mitochondrial carriers of the SLC25 family in the metabolism of plants.
There are some incorrect or confusing statements in the review that need correction, but this is an excellent review overall.
-Figure 1. This figure is difficult to interpret. Where are the other carriers?
Were membrane protein specific scoring matrices used? Were the residues of membrane directed parts included in the analysis? This is likely to give false signal as there is reduced evolutionary pressure on hydrophobic residues.
-Line 14
As a compartment of the central carbon metabolism, the mitochondria must be connected to the cytosol
>
As compartments of the central carbon metabolism, mitochondria must be connected to the cytosol
-Line 19
physiological functions whenever available > physiological functions, wherever available
-Line 23
we surveyed > we survey (to be consistent)
-Line 40
is a more stringent > a highly stringent
-Line 41
This highly impermeable nature > The highly impermeable
-Line 52
short helical region that is in parallel to the lipid bilayer > short helical region that is oriented parallel to the lipid bilayer.
-Line 54 Furthermore, each repeat region bears the structural motif PX[D/E]XX[K/R]X[K/R] (20–30 residues) [D/E]GXXXX[W/Y/F][K/R]G (IPR00193), and two hydrophobic transmembrane segments are connected by a long hydrophilic matrix loop [3,17].
The structural motif is incomplete and very dated, as it lacks for instance, the GXXXG motif, the Q brace, the PiXXXPi motif, the [FY]XX[YF] motif, Y brace and the cytoplasmic network residues. A correct sequence analysis of all of the conserved motifs is in Robinson, Overy and Kunji, PNAS 2008 and their role is explained in the recent structural papers: Ruprecht et al, PNAS 2014 and Ruprecht et al, Cell, 2019.
-Line 56 Our current understanding of MCFs is mostly derived from crystal structures of different versions and conformations of ADP/ATP carriers, which import ADP from the cytosol and export ATP from the MM [18]. Technically not correct, as there are also structures of the regulatory domains of the mitochondrial ATP/Pi carrier and aspartate/glutamate carrier (Yang et al 2014, Harborne et al 2015, 2017, Thangaratnarajah et al, 2014).
Line 58, better to cite the original structure papers (pebay-peyroula et al 2003, Ruprecht et al 2014, Ruprecht et al. 2019) than the review.
-Line 67 The transport cycle of MCFs was recently reviewed by Ruprecht and Kunji (2019) [21].
This review is about the ADP/ATP carrier, but it was extended to the other carriers in:
Ruprecht and Kunji, TIBS, 2019.
-Line 100: Recent work on the Arabidopsis TCA cycle interactome shows that a putative phosphate transporter interacted with TCA cycle enzymes [24,25]. However, to date, the physiological significance of these protein-protein interactions remains unknown.
Actually, this is unlikely to be true. The conformational changes of the carriers are such that there is no possibility of stable interactions with other proteins. Likely to be an artefact due the highly unstable nature of these proteins.
-Line 139 Import of pyruvate may be exclusively attributed to the mitochondrial pyruvate carrier (MPC) protein family.
No citations for this statement.
-Line 185 DICs are members of the MCF that facilitate the transport of dicarboxylates such as malate and related compounds across the IMM.
They can also transport phosphate and sulfate and thiosulfate, which are not related.
-Line 391 Our current understanding of MCFs is mostly due to work done on the yeast ADP/ATP Carrier > Our current understanding of MCFs is mostly due to work done on the yeast, fungal and mammalian ADP/ATP carriers (cite REF 18 and 19).
-Line 439: In this approach the candidate gene is cloned and homologously or heterologously expressed in either E. coli, Lactocossus lactis or yeast.
> In this approach the candidate gene is cloned and homologously or heterologously expressed in either E. coli, Lactococcus lactis or yeast.
Misspelling plus not italic. Also provide relevant refs of good examples.
-Line 448
Most MCF research groups circumvent this obstacle by co-incubating the radioactive substrate with the unlabelled candidate substrate and measure the inhibitory effect compared to the assay measurements without the competing substrate.
> Most MCF research groups circumvent this obstacle by co-incubating the radioactive substrate with the unlabelled candidate substrate and by measuring the inhibitory effect compared to the assay measurements without the competing substrate.
This assay only shows competition of the compound for the binding site, but does not prove transport. The better assay is to load the membrane vesicles/proteoliposomes with unlabelled compounds and monitor the uptake of the hot, which proves that the carrier exports the compounds.
-Line 469 Recent work on yeast mitochondrial pyruvate carriers (MPCs), a non-canonical MC, showed that mimicking the physiological pH gradient between the mitochondria and the cytosol resulted in quantifiable pyruvate transport. In the absence of the pH gradient no transport can be detected [75]. Furthermore, they have reported that the activity of MPCs are dependent on the formation of heterocomplexes.
The MPC belongs to a completely different protein family than the MC (SLC54 vs SLC25), and have no evolutionary relationship. They consist of two proteins, which have three TM helices each, and form a heterodimer.
-Line 473 and further:
This could also be true for members of MCF members, as is the case for Drosophila
DIC homologs [76]. Perhaps the transporter activity of the MCFs proteins can also be modulated by forming complexes with interacting partner/s. The glutathione transport function of OGC could not be replicated using the typical expression-purification-reconstitution-assay approach. An antiapoptotic protein, Bcl-2, was found to be an interacting protein partner of rat OGC and when co-expressed with OGC in CHO cells, the total mitochondrial glutathione content was significantly increased 24 h post-transfection [77].
There is no convincing evidence that MCF members form complexes. Claims in the past have been technical artefacts (see Crichton and Kunji, 2012). The conformational changes are so pronounced (see Ruprecht et al 2019) that there is no possibility of a stable interaction interface. Far more likely that the approaches yielded dysfunctional protein.
-Legend Figure 3 best to describe the abbreviations, as they come later in the review
-Ref 56 no space between from and Arabidopsis.
-Ref 64 ADP and ATP are in small case.
Reviewer 2 Report
The review "Plant Mitochondrial Carriers: Molecular Gatekeepers that Help to Regulate Plant Central Carbon Metabolism" is very interesting, considering that the exchange of metabolites between mitochondria and cytosol in plants is still unclear.
I suggest the authors to improve the text by editing the References (e.g. many terms are in lowercase and some species' names are not in Italics) and to correct some other minor details.
In particular:
line 64: delete "the" after "...is ready to begin a new";
line 468: "mitochondrial pyruvate carriers (MPCs), a (?) non canonical MC":
Finally, in lines 386-390, the abbreviations ADNT1 and APCs should be explained in the text. The codes for the proteins in this paragraph (and in lines 342-343) should be in in uppercase (like what shown in lines 193-194).
Reviewer 3 Report
In the review “Plant Mitochondrial carriers: molecular gatekeepers that help to regulate plant central metabolism” M. R. Toleco et al. focused on the plant mitochondrial carriers (MCs) involved in the transport of TCA cycle intermediates and in transport processes that are linked to oxidative phosphorylation either directly or indirectly. The authors reviewed the information available from studies in vitro and in planta; investigated the evolutionary relationship among dicarboxylate (DICs), dicarboxylate-tricarboxylate (DTCs), succinate-fumarate (SFCs) carriers and other non-plant mitochondrial organic acid transporters; examined old and new approaches to investigate MCs; and proposed some novel attractive areas of research to be addressed for understanding the physiological importance of MCs in the regulation of plant metabolism. The review is well written, interesting for a series of original remarks, conclusions and hypotheses, instructive and useful to a large spectrum of readers. It is therefore recommended for publication provided that the comments listed below are dealt with properly.
Lines 35-36: “inner mitochondrial space”? It should be “intermembrane space”.
Line 51: “and both the N and C termini face the intermembrane space.“ This relevant information about the topology and folding of MCs in the inner mitochondrial membrane requires a reference. The authors could choose one of the following references: Capobianco et al. Biochemistry 1991, 30, 4963-4969; Bisaccia et al. Biochemistry 1994, 33, 3705-3713; and Capobianco et al. FEBS Lett. 1995, 357, 297-300.
Lines 56-64: the first three lines of this paragraph should be reworded. Firstly, what was actually derived (actually, at least in part, confirmed – see below) from crystal structures of ADP/ATP carriers, is knowledge about the structure and the transport mechanism of MCs. In fact, plenty of information is available on the biochemical function and the physiological role of other MCs in different organisms. Therefore, in the first sentence “Our current understanding of MCs” should be changed in “Our current understanding of the structure of MCs”. Secondly, as far as the transport mechanism is concerned, it should be acknowledged that i) the transport cycle mechanism for AAC and other MCs (now called “alternating access mechanism”, see line 58) had been reported earlier with different names (“single site gated pore mechanism” by Klingenberg Trends Biochem. Sci. 1979, 4, 249–252 and Klingenberg BBA 2008,1778, 1978-2021; and “ping-pong mechanism” by Indiveri et al. BBA 1994, 1189, 65–73) and had been described before (ref. 11 and Palmieri and Pierri Essays Biochem. 2010, 47, 37-52) exactly as the authors of this review did in lines 59-64, and ii) in the earlier papers by Klingenberg and by Indiveri et al. the same transport mechanism was substantiated by functional and kinetic (not structural) evidence.
The authors seem to know that the transport mechanism described in lines 59-64 was older than the ref. quoted in line 58 (ref. 18) because at the end of their description they quote a paper of 2011 (see line 63) and stated that the mechanism “is supported by the crystal structure…” (line 65), which is perfectly right.
Line 63: in support of their description of the AAC transport mechanism the authors cite a recent paper (ref. 20) based on measurements of the free energy profile of AAC. However, they forgot to cite a very similar paper published one year before in the same journal (see Pietropaolo et al. BBA Bioenergetics 2016, 1857, 772-781). The authors should cite both papers or none.
Lines 211-213: the authors should perhaps give credit to the people who first measured the Km values of DIC and AAC, independently from the organism these carriers were from. Interestingly, the Km values of DIC and AAC, measured using both isolated mitochondria and purified proteins from various organisms, are very similar to the values reported for AtDIC and AtAAC in lines 211-213. Actually this observation is strongly in favour of the authors’ hypotesis mentioned in lines 213-214.
Line 213: “nanomolar range”? It should be “micromolar range”.
Line 242: “However”?
Line 247: “NtDIC1-4” or “NtDTC1-4” (as in line 257)?
Lines 285-287: why not quoting ref. 44 together with refs. 47-48 (in line 285) and ref 47 (in line 287)?
Line 274: “SCF” should be changed to “SFC”
Figs. 2A and 2C: it is suggested to change the order of the carrier catalysing 2-OG efflux as follows: DTC/SFC/DIC accordingly to the efficiency these carriers transport 2-OG.
Lines 312-314: an appropriate review strictly related to MCs and the sentence reported in these lines is: Palmieri and Pierri Essays Biochem. 2010, 47, 37-52.
Figs. 3A and 3B: in both figures NADH+ should be NAD+.
Lines 350-351: this sentence should be changed accordingly to the following two references (that dealt with AtPXN before ref. 54): Bernhardt et al. The Plant J. 2012, 69, 1-13; and Agrimi et al. Journal of Bioenergetics and Biomembranes 2012, 44, 333-340.
Line 359: “UCPs” should be changed “UCP1”, the first and only UCP showing protonophoretic function in brown fat.
Lines 360-361: the information that Arabidopsid was thought to have six UCP proteins (PUMP1-6 or UCP1-6) is missing. In phylogenetic trees of MCs from Arabidopsis and from Arabidopsis and mammals all UCP1-6 cluster together.
Lines 377-384: the interesting issue of the aspartate/glutamate transport orientation by UCP1-2 should be addressed here. Similarly the authors are suggested to discuss the possible role of dicarboxylate transport by UCP1-2 in this paragraph.
Lines 391-392: this sentence should be changed (see also above). It is apodictic and not fair to the literature and all the work done, for example, by Klingenberg and his group on the mammalian ADP/ATP carrier. Of note, the first 3D structure to be solved was that of the ADP/ATP carrier from bovine (not yeast).
Lines 401-411: another relevant paper on the function of Arabidopsis and mammalian APCs should be mentioned here (Monné et al. Journal of Bioenergetics and Biomembranes 2017, 49, 369–380). This paper shows that Arabidopsis APCs and their catalytic domains (APCs lacking their N-terminal domains containing the Ca2+-binding motifs) are able to transport Ca2+ and other divalent cations in complex with ATP. It is suggested to cite the above mentioned paper in lines 401, 405 and 408, and modify the text accordingly, especially in relation to the list of substrates accepted by APCs and their Ca2+-dependency.
Line 419: it is not clear what does the expression “they tend to accumulate …” refer to.
Lines 437-478: it is suggested to introduce the acronym “(EPRA)” of “the expression-purification-reconstitution assay” (see ref. 17) the first time it is mentioned (i. e. in line 438) and then use it in the rest of section 4.
Lines 451-454: the new approach that combines liposome reconstitution and mass spectrometry has also been applied to MCs (see the reference mentioned above: Monné et al. Journal of Bioenergetics and Biomembranes 2017, 49, 369–38).
Line 480: to help the readers references 11 and 17 should be introduced after the words “many plant MCs have been characterized in vitro”.
Lines 493-494: it could be interesting to let the readers know that the activity of the carnitine-acylcarnitine carrier (CAC) is also modulated by glutathionylation (Giangregorio et al., BBA 2013, 1830, 5299-5304).
